**PLOS** | NEGLECTED TROPICAL DISEASES

# *"Buruli ulcer and leprosy, they are intertwined"*: Patient experiences of integrated case management of skin neglected tropical diseases in Liberia

**Mateo Prochazka**[1]*, **Joseph Timothy**[1], **Rachel Pullan**[1], **Karsor Kollie**[2], **Emerson Rogers**[2], **Abednego Wright**[2], **Jennifer Palmer**[3]

**1** Department of Disease Control, Faculty of Infectious and Tropical Diseases, London School of Hygiene & Tropical Medicine, London, United Kingdom, **2** Liberia Ministry of Health and Social Welfare, Monrovia, Liberia, **3** Department of Global Health and Development, Faculty of Public Health and Policy, London School of Hygiene & Tropical Medicine, London, United Kingdom

* mateoprochazka@gmail.com

**Data Availability Statement:** The data supporting the findings of this study/publication are retained at the London School of Hygiene and Tropical

## Abstract

### Background

Skin neglected tropical diseases (NTDs) such as Buruli ulcer (BU) and leprosy produce significant stigma and disability. Shared clinical presentations and needs for care present opportunities for integrated case management in co-endemic areas. As global policies are translated into local integrated services, there remains a need to monitor what new configurations of care emerge and how individuals experience them.

### Methods

To explore patient experiences of integrated case management for skin NTDs, in 2018, we conducted a field-based qualitative case series in a leprosy rehabilitation centre in Ganta, Liberia where BU services were recently introduced. Twenty patients with BU (n = 10) and leprosy (n = 10) participated in in-depth interviews that incorporated photography methods. We contextualised our findings with field observations and unstructured interviews with health workers.

### Findings

The integration of care for BU and leprosy prompted new conceptualisations of these diseases and experiences of NTD stigma. Some patients felt anxiety about using services because they feared being infected with the other disease. Other patients viewed the two diseases as 'intertwined': related manifestations of the same condition. Configurations of inter-disease stigma due to fear of transmission were buffered by joint health education sessions which also appeared to facilitate social support between patients in the facility. For both diseases, medication and wound care were viewed as the cornerstones of care and appreciated as interventions that led to rehabilitation of the whole patient group through

Medicine and will not be made openly accessible due to ethical and privacy concerns. Data can however be made available after approval of a motivated and written request to the London School of Hygiene and Tropical Medicine at researchdatamanagement@lshtm.ac.uk.

**Funding:** This study was funded by the Accelerating Integrated Management (AIM) Initiative, a programme of American Leprosy Missions. The opinions, results and conclusions reported in this paper are those of the authors and are independent from the funding sources. The funders had no role in study design, data collection and analysis, decision to publish, or preparation of the manuscript.

**Competing interests:** The authors declare that they have no competing interests.

shared experiences of healing, avoidance of physical deformities and stigma reduction. Patient accounts of intense pain during wound care for BU and inability of staff to manage severe complications, however, exposed some shortcomings of medical care for the newly integrated service, as did patient fears of long-lasting disability due to lack of physiotherapy services.

## Significance

Under integrated care policies, the possibility of new discourses about skin NTD identities emerging along with new configurations of stigma may have unanticipated consequences for patients' experiences of case management. The social experience of integrated medication and wound dressing has the potential to link patients within a single, supportive patient community. Control programmes with resource constraints should anticipate potential challenges of integrating care, including the need to ameliorate lasting disability and provide adequate clinical management of severe BU cases.

## Author summary

Buruli ulcer and leprosy are disfiguring diseases of the skin that lead to disability and stigma. Although they are both present in overlapping population groups in West Africa, efforts to care for people affected have usually had a single disease focus. Since both conditions require wound care and long courses of medications, and the risk for transmission is minimal, it has been recommended that they could be managed together. However, there are few reports on how people affected by these diseases experience being brought together.At the border between Liberia and Guinea, a rehabilitation centre has recently integrated care for Buruli ulcer with leprosy care. We conducted a qualitative study in this facility to explore experiences of integrated care for these two diseases. We found that people experience these diseases in a similar way, to the extent that some considered both diseases to be related, and perceive receiving their medications and wound care together favourably.We identified some negative experiences including unjustified fear of acquiring the other disease, intense pain during the process of wound cleaning, and permanent disability due to lack of physiotherapy. In the future, programmes that implement integrated care should anticipate and address these negative experiences to mitigate the burden of these diseases.

## Introduction

Skin neglected tropical diseases (NTDs) are a group of stigmatising and disability-inducing infections that include leprosy, Buruli ulcer (BU), cutaneous leishmaniasis, lymphatic filariasis, onchocerciasis, mycetoma, and yaws. These diseases present as chronic disruptions and alterations of the skin barrier that cause significant scarring, burdening the lives of those affected through reduced mobility, hindered productivity and psychological distress[1–4].

As highly visible diseases that affect a person's aesthetic appearance, skin NTDs produce social effects including stigma. Stigma is the social disqualification of individuals due to an attribute that marks them as different from "normal" and links them to undesirable characteristics[5,6]. Stigma may manifest as active experiences of social judgement and discrimination

("enacted stigma"), or as an internalised sense of shame and fear of encountering enacted stigma ("felt stigma")[7]. As infectious diseases, skin NTDs can also generate stigma based on an exaggerated fear of danger and contagion[8]. While what is stigmatised and the forms that stigma takes vary between cultures and conditions, an emerging literature on skin NTDs increasingly recognises that stigma associated with these diseases inevitably shapes people's health and health-seeking behaviour and the effectiveness of disease control[6–9].

Although skin NTDs have different aetiologies and transmission routes, they share a need for medically complex and multi-dimensional case management. This can include long courses of chemotherapy, in-patient wound care, prevention of physical disability and psycho-social support to achieve rehabilitation[4]. Traditionally, all skin NTDs have been managed through single-disease vertical programmes that duplicate efforts in some areas and suffer similar gaps in investment in others, resulting in fragmented and ineffective responses[10]. These shared needs represent opportunities for integrated case management strategies that, if implemented effectively, may contribute to reducing the burden of these diseases[11–14].

Recently, these opportunities have been translated into global policy recommendations for integrated control programmes that aim to simultaneously address all co-endemic diseases while increasing efficiency over stand-alone programmes[4]. Arguments in support of this integrated approach include improving resource allocation, developing sustainable and flexible expertise, and leveraging existing regional responses to co-endemic diseases[12,15]. As global policies are translated into local integrated services, however, there remains a need to monitor what new configurations of care emerge and how individuals receive them. Such insight would ensure people's experiences are kept at the forefront of programming, and provide depth and richness about the context in which local NTD control programmes operate[16,17].

Liberia is endemic for several skin NTDs, including BU and leprosy. Emerging evidence suggests that in this setting both diseases disrupt people's lives through chronic illness, morbidity and disability[18]. Towards establishing a centre of excellence for the management of NTDs, the Ministry of Health has recently integrated BU care at a national leprosy rehabilitation centre in Ganta, a town close to the northern border with Guinea[19]. The aim of this qualitative study was to explore the experiences of in-patients affected by BU and leprosy receiving integrated case management in this setting.

## Methods

We conducted a field-based qualitative case series of patients with BU and leprosy receiving care at the Ganta Tuberculosis & Leprosy Rehabilitation Centre in Nimba County, Liberia. This methodological approach was chosen on the basis of conducting patient-centred research in a clinical context. We used in-depth interviews with cases, daily field notes and photography methods during six weeks of fieldwork between June and August 2018.

### Study setting and population

The centre is a national referral facility that provides diagnostics, treatment, and management for people of all ages with leprosy, BU, or tuberculosis. Established in the mid-1920s by American Methodist missionaries, the centre initially provided rehabilitation services for people affected by leprosy, subsequently leading to a colony of people with leprosy in the surrounding area. The centre incorporated tuberculosis care in the early 1990s during the First Liberian Civil War. Based on WHO-driven initiatives to integrate case management for skin NTDs, the Ministry of Health decided to upgrade the centre's capacity, incorporating BU care with leprosy care since 2015. Following the National Strategic Plan for the Case Management of NTDs, centre staff was trained in laboratory diagnostics and care for BU[19].

Members of the community or health workers refer potential patients affected by BU or leprosy to the centre on the basis of clinical features. Suspected cases are confirmed upon arrival at the centre using acid-fast bacilli (AFB) smears for both diseases. Most patients with BU and leprosy are managed as in-patients and receive long-term care. Both patient groups cohabit and receive care as a single group with shared facilities and health workers. Patients with sufficient mobility might be managed as outpatients if they are able to commute daily for case management activities. Lodging and treatment for patients with tuberculosis occurs in a separate building.

At the centre, patients with multi-bacillary leprosy, defined as having at least 5 skin lesions, receive one year of multidrug therapy (MDT) with rifampicin, clofazimine and dapsone. People with pauci-bacillary leprosy (<5 lesions) receive MDT for six months. People with BU receive 56 days of rifampicin plus clarithromycin or streptomycin, according to availability. Both patient groups receive daily wound care for their lesions which include extensive skin ulcers potentially leading to scarring and contractures in BU and neuropathic ulcers in leprosy [4]. Other skin manifestations of leprosy such as skin patches and nodules do not require wound care. All laboratory procedures, care activities, lodging and food provision are offered free of charge to patients; funding is sourced from a variety of external donors including the German Leprosy and Relief Association (GLRA), and coordinated via the Ministry of Health. During the time of the fieldwork, approximately 90 patients with BU and 30 patients with leprosy were receiving care at the centre.

### In-depth interviews

We sought 20 patients ("cases") with a diagnosis of BU (n = 10) or leprosy (n = 10). We included patients who had received care at the centre for at least two weeks to ensure they had experienced case management activities. We excluded patients with cognitive impairment that could make interviewing difficult, and underage (<18 years) patients without the presence of a guardian since they could not legally provide written informed consent.

During the first week of field work, potential participants were selected by a researcher (MP) during informal conversations or via clinical chart review to identify patients that met the inclusion criteria. We invited patients to participate in the study during delivery of chemotherapy and wound care services, or other activities in common areas in the centre. We purposively sampled participants across the range of ages, genders, languages and counties in the sample to ensure maximum variability of experiences. Health workers did not have a role in selecting potential study participants.

Interviews were semi-structured and focused on eliciting patients' experiences of illness and health-seeking before admission to the hospital, as well as experiences of case management and rehabilitation after admission. During the first week of fieldwork, preliminary question guidelines were validated with a subset of inpatients (n = 4) not included in the final study dataset. We continued to refine interview guidelines during data collection to include additional probing on emerging themes. Interviews took place in a private office at the centre and lasted between 38 and 60 minutes. All participants were interviewed in English by the first author, with support of translators for local languages (Bere, Gio and Mano). To obtain rich data, interviewing was conducted using respondent-centred approaches, applying interview guidelines flexibly while allowing participants to take control of the conversation and using probing techniques after their answers. All interviews were audio recorded. Information on the gender, age, and diagnosis of cases was extracted from clinical charts.

## Field notes

The lead researcher's observations, perceptions, and experiences of interacting with patients and staff were collected as daily field notes or recorded as voice memos, and transcribed at the end of each day. Unstructured conversations with five staff members (two nurses, one doctor, one laboratory technician and one programme manager) were also included to contextualise the content of in-depth interviews with patients.

## Photography

The study involved the production of photographs as primary data. Using an instant camera (Fujifilm Instax Wide), the case management setting, activities and routine procedures, including lesions of patients, were documented during the study period. These photographs did not include patients' faces or identifying features. Additionally, photographs from atlases of leprosy[20] and BU[21] were used to depict different categories of wounds. During in-depth interviews, both types of photos were integrated into the interview guideline and shown to the study participants as physical prints. Discussions on what they saw, thought, and experienced provided additional meaning to concepts elicited through verbal questions.

## Data management and analysis

Audio recordings from all in-depth interviews were transcribed using Microsoft Word 2016 (Version 16.13) with support from a multi-lingual research assistant. English portions of the interviews were transcribed verbatim. Thematic analysis occurred simultaneously with data collection. After six interviews, two clean transcripts were uploaded to Dedoose version 8.0, a software for qualitative analysis. We used a first cycle of coding to develop a tentative codebook and identify emerging themes, using primarily an inductive approach. Deductive coding based on a literature review of patient medical and social experiences of skin NTDs from other settings also contributed to initial codebook development. This codebook served as a provisional start list to develop a case-theme matrix in Microsoft Excel 2016 (Version 16.16.15). The matrix included rows for all cases and columns for each tentative theme, facilitating within-case and across-case analysis. During fieldwork, excerpts from interview transcripts were entered into the matrix in the appropriate case vs. theme cell using pattern coding. Matrix cells also contained descriptive and analytic memos concerning the specific theme in the context of each specific case. The matrix was rearranged and recompleted iteratively throughout the analysis process, as new themes emerged and others converged into larger themes. Thematic saturation for main themes was reached by case 16, four additional cases were included aiming for maximum variability in experiences. Themes were condensed into narrative descriptions of findings. We reported our findings as patient experiences to avoid strictly biomedical interpretations that might further remove future programming from users' needs. While other approaches, such as emphasising experiences of health workers and operational challenges for programming are also needed, trustworthiness in the exploration of socio-cultural themes such as stigma, disability, and rehabilitation required a patient-centred approach[17]. Representative quotes were embedded within the results to exemplify themes, with minor grammatical alterations to improve readability. Field notes and primary photographs were used to complement descriptions and provide additional context[22]. Case vignettes and descriptions were summarized in Table 1; clinical severity and disability classifications were reported according to WHO guidelines[21].

**Table 1.** Case description and vignettes.

| N° | DIAGNOSIS | SEX | AGE | OCCUPATION | CASE DESCRIPTION* | CASE MANAGEMENT** | COUNTY | LANGUAGE |
|---|---|---|---|---|---|---|---|---|
| 01 | BU–Cat. III | F | 36 | Business | Two months of nodule with oedema and small ulcers in right leg. | Two weeks at the centre. Receiving rifampicin + clarithromycin (56 days). Daily wound care. | Maryland | English |
| 02 | BU–Cat. III | M | 22 | Driving student | One month history of developing six large ulcers in right leg. | One year at the centre. Completed rifampicin + streptomycin (56 days). Daily wound care. | Dieke, Guinea | English |
| 03 | MBL–G1D | M | 37 | Judiciary | Three years of plaques, nodules and home-managed ulcer in limbs. | Six weeks at the centre. Receiving WHO-MDT. No wound care. | Nimba | English |
| 04 | MBL–G1D | M | 28 | Cocoa farmer | One year of numbness and skin nodules. | Four months at the centre. Receiving WHO-MDT. No wound care. | Nimba | English |
| 05 | BU–Cat III. | F | 68 | Peanut farmer | One week of ulcer in right leg. | Two months at the centre. Receiving rifampicin + clarithromycin (56 days). Daily wound care. | Lofa | Gio |
| 06 | MBL–G0D | M | 21 | High-school student | One and a half year of skin patches and small ulcer. | Two weeks at the centre. Receiving WHO-MDT. Received daily wound care. | Nimba | English |
| 07 | BU–Cat II. | M | 22 | High-school graduate | Five months of ulcerated plaque in left leg. | Three weeks at the centre. Receiving rifampicin + streptomycin (56 days). Daily wound care. | Bong | English |
| 08 | MBL–G2D | F | 55 | Rice farmer | Two years of peripheral neuropathy in left arm. | Two months at the centre. Receiving WHO-MDT. No wound care. | Nimba | Mano |
| 09 | MBL–G2D | F | 66 | Farmer | Four years of peripheral neuropathy and skin patches. | One year at the centre. Completed WHO-MDT. No wound care. Leprosy reaction management with steroids. | Nimba | Mano |
| 10 | MBL–G2D | F | 20 | Farmer | Seven years of peripheral neuropathy. Ulcer with osteomyelitis on left foot. | Seven months at the centre. Surgery for complicated ulcer. Daily wound care. | Lofa | English |
| 11 | BU–Cat III. | M | 47 | Construction engineer | Three days of oedema and ulcer in right leg. | Three months at the centre. Receiving second course of rifampicin + clarithromycin (56 days). Weekly wound care as outpatient, self-manages wounds at home. | Nimba | English |
| 12 | BU–Cat III. | M | 42 | Farmer | Two weeks of extensive swelling and ulcer in left arm. | Six months at the centre. Completed rifampicin + clarithromycin (56 days). Daily wound care. | Nimba | English |
| 13 | BU–Cat III. | F | 51 | Business | One year of oedema and ulcers in both legs. | Two weeks at the centre. Receiving rifampicin + clarithromycin (56 days). Daily wound care. | Gbarpolu | English |
| 14 | BU–Cat III. | M | 15* | Student | Six months of extensive ulcers in both legs, severe malnutrition and sepsis. | Two months at the centre. Completed rifampicin + clarithromycin (56 days). Daily wound care. *Died on 26/07/18.* | Bong | Bere |

(*Continued*)

**Table 1.** (Continued)

| N° | DIAGNOSIS | SEX | AGE | OCCUPATION | CASE DESCRIPTION* | CASE MANAGEMENT** | COUNTY | LANGUAGE |
|---|---|---|---|---|---|---|---|---|
| 15 | MBL–G2D | M | 46 | Security | Two years of skin patches and numbness in limbs. | Two months at the centre. Receiving WHO-MDT. No wound care. | Sinoe | English |
| 16 | MBL–G2D | F | 55 | Farmer | One year of numbness and weakness in hands and feet, skin nodules and patches. | Seven months at the centre. Receiving WHO-MDT. No wound care. | Nimba | Gio |
| 17 | MBL–G2D | M | 42 | Miner | Five years of numbness in feet and skin patches | Three months at the centre. Completed WHO-MDT. Daily wound care. | River Gee | English |
| 18 | BU–Cat III. | F | 60 | Farmer | Recurring episodes of oedema in both legs since 2014, blister and ulcer in left leg for three weeks. | Two months at the centre. Completed rifampicin + clarithromycin (56 days). Daily wound care. | Nimba | Gio |
| 19 | BU–Cat III. | M | 17 | Student | Two months of oedema, blisters and ulcers in both legs. | Six months at the centre. Completed rifampicin + clarithromycin (56 days). Daily wound care. | Monrovia | English |
| 20 | MBL–G2D | F | 47 | Business | Few days of oedema and skin nodules in face and limbs. | Completed WHO-MDT as in-patient. Receiving corticosteroid therapy for leprosy reaction type II. | Bong | English |

**NOTES:** BU = Buruli ulcer, MBL = Multibacillary leprosy, Cat = Category, G0D = Grade 0 Disability, G1D = Grade 1 Disability, G2D = Grade 2 Disability, WHO-MDT = WHO Multidrug therapy

*At presentation in the centre

**At time of interview

## Ethics

This study was approved by the Ethics Committees of the London School of Hygiene and Tropical Medicine (15584) and the Liberian Ministry of Health (1802088). Those who expressed interest to participate gave their written informed consent. As most participants were illiterate, a research assistant ensured that participants understood the study procedures thoroughly after a verbal explanation in the patients' mother tongue. Patients who were under 18 years gave verbal informed assent, and were interviewed in the presence of guardians, who gave written informed consent. Participants received a towel, a piece of soap, and a photograph of themselves, equivalent to less than 2.00 USD as compensation for their time. All photographs were taken with the verbal consent of those depicted, including staff, and were shown to them immediately, receiving favourable responses. Due to the stigmatising nature of BU and leprosy, all photographs were produced in the ethical framework of representing vulnerable subjects[23].

## Results

We interviewed 20 people affected by BU (n = 10) and leprosy (n = 10); participant demographic and clinical characteristics are described in Table 1. Additionally, we entered 28 daily entries as field notes, which included unstructured interviews with five health care workers. Our results are presented in three sections which correspond to patient experiences of three dimensions of care provided by the NTD programme: (1) health education, disease identities and inter-disease stigma, (2) experiences of pain and healing through wound care and medication, and (3) experiences of rehabilitation addressing stigma and disability.

### "Intertwining" disease identities and inter-disease stigma

Due to its history, the Centre and its surroundings had a negative reputation as a place of leprosy, disability, and amputation. Participants mentioned that members of the centre's surrounding communities were sometimes labelled as "leprosy people", which made others scared of approaching the facility. The recent inclusion of BU care within a well-known leprosy rehabilitation facility caused some apprehension among newcomers with BU, who anticipated leprosy-related stigma:

"I was thinking that this place was only for leprosy, not for sores [ulcers] like this [. . .] I said, so when I come, they will not cut [amputate] my foot?" (Case 013, BU, F, 51)

Some participants appeared unable to distinguish between the aetiologies of skin ulcers from the two diseases, or felt distinguishing them was unnecessary. They perceived both conditions to be "intertwined" or related to one another along a continuum of the same disease, suggesting that the integration of care had created new dimensions in old discourses about leprosy:

"Buruli ulcer and leprosy, they are intertwined. [. . .] When the Buruli ulcer, it stays long on you, I think leprosy will join into it. Then your foot and other things will be gone [amputated], so they are intertwined. Because it is sore and caused by bacteria and diseases." (Case 007, BU, M, 22)

Other participants perceived the diseases as separate conditions but with similar attributes. Leprosy was perceived to be more severe because of an expectation of amputation and social discrimination. Such distinctions helped construct social identities that shaped notions of otherness among patients with the different conditions within shared spaces:

"I have my infection, they have theirs, ok?" (Case 015, MBL, M, 46)

These notions of "them" and "us" shaped inter-disease stigma among both patient groups within the centre, manifested as fear of acquiring the other disease. Some people affected by BU reported "feeling sorry" for people with leprosy, and feared developing leprosy themselves. This sometimes motivated them to use outpatient rather than the integrated inpatient care unit, and made them avoid common bathrooms and mosquito bites in the centre, both thought to facilitate leprosy transmission:

"The reason [I stay home] is that I have Buruli. You have leprosy, then you laying down as a leprosy patient, then I am laying down as a Buruli patient. Mosquito, is able to bite you, and come and bites me. So, the fear is that the person who has leprosy and the person who has ulcer can't be in one place. I am afraid." (Case 011, BU, M, 47)

Stigma was bidirectional, as newcomers with leprosy also avoided people in early stages of BU treatment due to the smell of the ulcers and fear of transmission:

"First I was afraid they were going to put me in a room with some of the patients that have the sore. I said to the people: I'm not sleeping there, I'm not gonna get the sore too." (Case 006, MBL, M, 21)

Health workers, however, pointed out that inter-disease stigma seemed to be buffered over time by morning health education sessions, where information about BU and leprosy transmissibility was frequently discussed. This knowledge was evident among some participants who described leprosy as not transmissible after patients receive medication, and that BU was transmissible by walking in swamps but not from person to person. Morning sessions also facilitated relationship-building among people with similar backgrounds but different diseases thereby encouraging communal living. This was apparent in the case of a man with BU who frequently spent time with a man affected by leprosy. Both young men were from Guinea and were learning English during their time at the centre. After asking for a portrait of them taken together, the man with BU described the photograph as follows:

> "This is me. And my Guinea man. Leprosy man. From Guinea. My friend." (Case 002, BU, M, 22).

Establishing an intertwined patient community helped people from both disease populations to support each other emotionally and instrumentally throughout integrated medical activities, discussed next.

## Medication and dressings: Experiences of integrated chemotherapy and wound care

Most participants were aware of differences in the types of medication and wound care required for each disease. Both patient groups experienced receiving daily medication as the cornerstone of their rehabilitation: tablets and pills were described as effective and appreciated for their ability to remove illness and its attached social burden. Medication inspired hope about a future without disease:

> "I feel comfortable, when there's life, there is hope. Because I am going through medication, and you know my time is not yet over." (Case 015, MBL, M, 46)

Though their one year course of medication was perceived as lengthy, people with leprosy framed completing it as necessary to restore their social identities:

> "I will say (to others in the community): I took the medicine and I don't get any sickness now. You must not be afraid of me, you come around me, the way we were joking before, it will be like that now." (Case 009, MBL, F, 66)

A few people with BU experienced only limited wound improvement and maintained positive AFB smears, requiring a second course of treatment with the same regimen. This caused some to doubt the treatment efficacy:

> "What I believe in my mind is that BU has no good treatment yet, they have not discovered it for this sickness." (Case 011, BU, M, 47)

Dressing of skin ulcers was provided simultaneously to all people affected by BU and a minority of those with leprosy. Discussions about this activity emotively described the enacted stigma people experienced associated with smell:

> "The way of the sore, and the smell of it, people don't want to sit around you. Yeah, they think they might get it, they think it's transferrable." (Case 006, MBL, M, 21)

Both groups perceived wound cleaning and dressing as healing by taking away germs, replacing foul smelling bandages, and avoiding flies and felt this helped people avoid stigma.

Attitudes towards wound care were also shaped by experiences of pain, which were prominent during the initial stages of wound cleaning. Upon admission, people with skin ulcers underwent manual debriding consisting of removal of necrotic tissue, fluid, and plasters of mud remaining from traditional medicine practices. Some participants interpreted the reduction of pain throughout the wound care process as evidence of improvement, or as a necessary experience to achieve cure, as discussed by a woman with BU while describing a photograph of the wound care room:

"In this place they can change the bandage, and this is the place I said that day that you must take my picture so it can be remembered, because I suffered under this thing [Buruli ulcer] too much [. . .] So I want you to give me the picture when I go, then I can show it to my children, in this place I was cured". (Case 013, BU, F, 51)

Patients with severe BU required squeezing and draining of subcutaneous accumulations of fluids. Pain management was limited to injection of intramuscular diclofenac, the strongest analgesic available. Instances of screaming, intense agitation, and fainting were observed during fieldwork and documented in participants' testimonies. Such experiences contributed to occasional refusal or resistance to participate in wound care activities. This was the case of a young man with BU frequently seen arguing with staff during wound cleaning. He described a photograph of patients queueing outside the wound care room as follows:

"The time we can be going for dressing our hearts can be beating, we [are] going [to] feel pain here. Any time we see this place, each time we know our hearts can be disturbed." (Case 019, BU, M, 17)

People with large wounds due to BU often arrived at the centre presenting with fever, infected wounds in need of antibiotics, and severely malnourished, especially after prolonged and unsuccessful experiences with traditional medicine, as expressed by the mother of a young boy with extensive BU lesions in both legs:

"That was four months we made there [spent at the herbalist], it was not working, he was just reducing the body, (. . .) losing weight." (Case 014, BU, M, 15)

Such severe cases required surgical management such as amputation, skin grafting or extensive debridement under anaesthesia, which, theoretically, could be coordinated via referral to a neighbouring hospital. However, staff complained that they were discouraged from organising referrals because of resistance from staff at receiving facilities who feared disease transmission or cited limited resources to cover surgical costs.

Complicated cases such as these thus challenged the centre's management capacity. The boy severely affected by BU described above died the day after participating in an interview due to sudden onset of diarrhoea and dehydration complicating his vulnerable state. Lack of emergency equipment, overnight medical staff, and capacity to establish a central venous catheter impaired adequate shock management. In the aftermath of the emergency, two health workers admitted that deaths from shock and complications among people severely affected by BU was a recurrent and particularly distressing problem.

## "Small, small": Progressive visible improvement, normalisation and rehabilitation

Notwithstanding instances such as the one described above, favourable experiences of case management dominated participants' narratives; these emphasised the external visibility of improvements and the reduction of felt stigma and the anxiety caused by anticipated stigma. Testimonies highlighted the noticeable closing of open sores in BU and disappearing of skin marks in leprosy as desirable outcomes of treatment. This was highlighted by a participant describing the experience of a family member who previously received early treatment for leprosy, and whose favourable results included resuming economic productivity and avoiding social isolation due to lack of discrediting scarring:

"A decent man, he's working! The whole fingers correct, the toes correct, everything on him is fine! No problem, you can't even know he had leprosy." (Case 006, MBL, M, 21)

Improvements in mobility and decreasing dependence on assistive devices such as wheelchairs, crutches and walking sticks were also emphasised in participant's narratives. Affected limb movement and function was facilitated by patient's own motivations to resume social and economic productivity, and by participants' fear that their bodies would be permanently disabled. Testimonies about the rehabilitation process also acknowledged deficits in the centre's programming. While assistive devices were available from a local workshop that facilitated makeshift wheelchairs and shoewear, formal physiotherapy services including guidance in rehabilitative exercising were not available due to understaffing. Weekly medical rounds focused on individual adherence to chemotherapy and continuous wound care, and tended to neglect physical rehabilitation plans to prevent lasting functional disability. This had severe consequences for some people affected by BU, for whom a lack of physiotherapy and guidance led to disfigurement potentially incompatible with productivity:

"I can't come back to my normal stage (. . .) The way I used to work with my two hands, it will not be like that way now. Maybe sometime, if I don't get some family to help me, to get some money, to make little business and then sit down. . . I will not be working hard again, that's the only thing I am thinking about, the work." (Case 012, BU, M, 42)

Overall, patients appreciated the long-term commitment to care that the programme and staff provided. Both groups of participants referred to these improvements using the phrase "small-small" to signify how cure occurred slowly, but surely:

"They started to dressing me every morning, give me medicine, they are doing well. At least I can see myself moving on my feet now small, small." (Case 001, BU, F, 36)

Case management at the centre was described as a rehabilitating and "normalising" experience that addressed both disability and stigma, as exemplified by a man with BU while discussing a photo of a patient waiting area:

"People who come here are people who are getting well [. . .] They [are] able to walk by themselves [. . .] They feel that they are getting normal, they are getting better." (Case 011, BU, M, 47)

Furthermore, bringing together both patient groups in a single facility for this treatment had the effect of blurring the lines between disease identities, not only shaping unique

configurations of inter-disease stigma, but also linking patient groups within a single patient community. This was verbalised by the mother of the boy with BU who died, who described the way integration of care in one facility intertwined disease identities in the collective imaginary:

> "The sore people and the leprosy people they are one, they are in one place." (Case 014, BU, M, 15)

## Discussion

In this qualitative study, we explored patients' experiences of integrated case management of BU and leprosy in a referral in-patient facility in Liberia. Our study is among the first to document patient experiences of multiple skin NTDs in a specialised clinical centre. Upon integration of care, disease identities of people with these skin NTDs intertwined, reshaping old discourses and configuring inter-disease stigma. Overall, case management activities appeared to adequately address the burden of both diseases, slowly rehabilitating people's "spoiled" social identities by tackling stigma and disability[24]. Moreover, the long-term, shared and occasionally intense social experiences of medication and wound dressing created a patient community that supported patients with both diseases. Negative experiences of integrating BU care in a leprosy facility with limited resources, however, included long-lasting disability due to lack of physiotherapy services, suboptimal pain management during wound care, and life-threatening complications among severe cases of BU. These experiences are expressions of actionable implementation challenges that should be considered in the design of people-centred approaches.

While integrated approaches to the case management of skin NTDs have been implemented in co-endemic areas, few health systems have documented actual experiences of implementation[25]. A case study in Ethiopia reported the managerial feasibility of integrating morbidity management of tropical lymphedema caused by lymphatic filariasis (LF) and podoconiosis[26]. Patient experiences of integrated care for leprosy and BU in Cote d'Ivoire[25], LF and leprosy in Nepal[27], and multiple skin NTDs in Liberia[18] have been explored in community settings, but studies investigating the social experience of integrated case management in facility settings are lacking. To our knowledge, our study is the first to examine patient experiences of integrated clinical case management of skin NTDs.

We found positive experiences of receiving integrated case management among people affected by BU and leprosy. Participants' testimonies unveiled common experiences of illness and shared needs for holistic care, including physical rehabilitation: these shared social experiences further enabled integration. While at a global level, the justification to integrate programmes for skin NTDs is primarily logistical or resource considerations[4,15], here, we demonstrate that shared experiences of morbidity, disability and stigma also facilitate integrated approaches to case management, and enhance their feasibility.

Our findings suggest that service integration might have an unanticipated effect on disease identities, leading to the intertwining of leprosy and BU in patient's discourses and inter-disease stigma. Based on these findings, we hypothesize that as skin NTD programmes transition towards integration, the way people perceive and experience these diseases might change, affecting stigma and potentially also health-seeking behaviour. While enacted and felt stigma have been extensively documented for skin NTDs[28,29], there is limited documentation of inter-disease stigma. This phenomenon had been anticipated in a community-based study in Nepal that explored attitudes towards integrating care among people with LF and leprosy,

finding perceptions of stigma concerning the alternate condition[30]. In agreement, our findings on inter-disease stigma reinforce that the integration of case management for skin NTDs should anticipate this phenomenon in mixed patient communities. While constant contact between both patient communities may eventually help reduce stigma[31], the experiences described at Ganta suggest that inter-disease stigma might also be actively addressed by health education sessions. These should be focused on mitigating fear of transmission, and delivered from the onset of integrated case management. Additionally, experiences of cross-disease community building suggest that interventions that leverage social support might also be effective in buffering both disease-specific and inter-disease stigma. Our findings support experiences from "therapeutic communities" as part of BU management in Benin[32], and highlight the need for psychosocial interventions in which patients cross-support each other in their path towards rehabilitation.

Our study identified negative experiences of case management in a resource-constrained setting, which include limited rehabilitation due to lack of physiotherapy services and experiences of intense pain and death among people with BU. Liberia's history of civil war, Ebola, and socio-political context shape a weakened health system in which effective programme implementation is challenging[33,34]. While broader health-system strengthening is needed, these particular implementation challenges can be directly targeted. People with severe BU might require complex management due to life-threatening and painful complications including secondary bacterial infections, sepsis, and shock[35]. Establishing protocols and algorithms for pain management, referral of complex cases, and shock management is urgently needed to avoid recurrence. Additionally, holistic approaches to case management that focus on social rehabilitation and not solely on biomedical or operational outcomes (e.g. negative smears or treatment completion) are needed to prevent lasting disability and stigma[12,36,37]. This should include streamlining quality rehabilitation services during in-patient care with follow up community-based rehabilitation upon discharge[38].

This study had some limitations. As a health facility-based study, people in the community who did not access care were not represented in the study sample. While understanding experiences of integrated case management implied recruiting inpatients, this approach might have excluded people who fear seeking care due to stigma and who might have experienced integrated case management differently. Future community-based studies on integrated case management for skin NTDs are needed. Additionally, the interviewer did not speak Gio, Bere, or Mano. Language, gender and cultural differences might have affected experiences shared due to positionality and social desirability. The use of photography and fluent translators well known in the facility might have bridged some of these barriers.

## Conclusion

Based on patient experiences, integrating case management of leprosy and BU in this context appears to be effective in addressing the needs of people affected by these diseases. Integration was experienced positively by patients, but reshaped discourses on disease identity around unique inter-disease stigma configurations. Patients experienced consequences of limited resources, such as lack of support to reduce lasting disability, and poor management of BU complications.

Our study further supports the role of applied social sciences in locating people's experiences at the forefront of programme design and implementation. Future social science research should study how stigma is shaped and performed under the new integration paradigm for the management of skin NTDs. Further patient-centred research and health system strengthening are needed to support the continued development of sustainable and coherent integrated care programmes for these diseases.

## Acknowledgments

We would like to thank John Brimah and all staff from the Ganta TB & Leprosy Rehabilitation centre for their ongoing support to conduct this research. Emmanuel Porkpah eagerly contributed to data collection and real time translation, making this study possible. Laura Dean from the Liverpool School of Tropical Medicine gave key suggestions prior to fieldwork which helped to anticipate potential challenges. Finally, we would like to thank patients at the centre and the surrounding communities in Nimba County for their welcoming and warm attitude.

## Author Contributions

**Conceptualization:** Mateo Prochazka, Joseph Timothy, Emerson Rogers, Jennifer Palmer.

**Formal analysis:** Mateo Prochazka.

**Funding acquisition:** Rachel Pullan.

**Investigation:** Emerson Rogers, Abednego Wright.

**Methodology:** Mateo Prochazka, Jennifer Palmer.

**Project administration:** Mateo Prochazka, Karsor Kollie, Emerson Rogers, Abednego Wright.

**Resources:** Rachel Pullan, Karsor Kollie, Emerson Rogers, Abednego Wright.

**Software:** Mateo Prochazka.

**Supervision:** Joseph Timothy, Abednego Wright, Jennifer Palmer.

**Writing – original draft:** Mateo Prochazka, Jennifer Palmer.

**Writing – review & editing:** Mateo Prochazka, Joseph Timothy, Rachel Pullan, Karsor Kollie, Jennifer Palmer.

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
