## [Decision Letter · Decision Letter 0]

18 Sep 2019

Dear Dr Prochazka:

Thank you very much for submitting your manuscript "“Buruli ulcer and leprosy, they are intertwined”: patient experiences of integrated case management of skin neglected tropical diseases in Liberia" (PNTD-D-19-01195) for review by PLOS Neglected Tropical Diseases. Your manuscript was fully evaluated at the editorial level and by independent peer reviewers. The reviewers appreciated the attention to an important topic but identified some aspects of the manuscript that should be improved.

We therefore ask you to modify the manuscript according to the review recommendations before we can consider your manuscript for acceptance. Your revisions should address the specific points made by each reviewer.

(1) A letter containing a detailed list of your responses to the review comments and a description of the changes you have made in the manuscript.

(2) Two versions of the manuscript: one with either highlights or tracked changes denoting where the text has been changed (uploaded as a "Revised Article with Changes Highlighted" file ); the other a clean version (uploaded as the article file).

(3) If available, a striking still image (a new image if one is available or an existing one from within your manuscript). If your manuscript is accepted for publication, this image may be featured on our website. Images should ideally be high resolution, eye-catching, single panel images; where one is available, please use 'add file' at the time of resubmission and select 'striking image' as the file type. 

Please provide a short caption, including credits, uploaded as a separate "Other" file. If your image is from someone other than yourself, please ensure that the artist has read and agreed to the terms and conditions of the Creative Commons Attribution License at http://journals.plos.org/plosntds/s/content-license (NOTE: we cannot publish copyrighted images). 

(4) Appropriate Figure Files 

Please remove all name and figure # text from your figure files upon submitting your revision. Please also take this time to check that your figures are of high resolution, which will improve both the editorial review process and help expedite your manuscript's publication should it be accepted. Please note that figures must have been originally created at 300dpi or higher. Do not manually increase the resolution of your files. For instructions on how to properly obtain high quality images, please review our Figure Guidelines, with examples at: http://journals.plos.org/plosntds/s/figures

While revising your submission, please upload your figure files to the Preflight Analysis and Conversion Engine (PACE) digital diagnostic tool, https://pacev2.apexcovantage.com/ PACE helps ensure that figures meet PLOS requirements. To use PACE, you must first register as a user. Then, login and navigate to the UPLOAD tab, where you will find detailed instructions on how to use the tool. If you encounter any issues or have any questions when using PACE, please email us at figures@plos.org.

We hope to receive your revised manuscript by Nov 17 2019 11:59PM. If you anticipate any delay in its return, we ask that you let us know the expected resubmission date by replying to this email.

To submit your revised files, please log in to https://www.editorialmanager.com/pntd/

Sincerely,

Uwem Friday Ekpo, PhD

Associate Editor

Gerd Pluschke

Deputy Editor

Reviewer's Responses to Questions

**Key Review Criteria Required for Acceptance?**

**Methods**

-Are the objectives of the study clearly articulated with a clear testable hypothesis stated?

-Is the study design appropriate to address the stated objectives?

-Is the population clearly described and appropriate for the hypothesis being tested?

-Is the sample size sufficient to ensure adequate power to address the hypothesis being tested?

-Were correct statistical analysis used to support conclusions?

-Are there concerns about ethical or regulatory requirements being met?

Reviewer #1: The research is largely focused on the experiences of the patients, and the authors have described this as a field-based qualitative case series. However, it is not clear the rationale for selecting this qualitative design approach. As there are several other qualitative approaches that could have been adopted, the authors may wish to state their reason(s) for this choice.

Although the authors have mentioned the overall aim of the study, ‘The aim of this qualitative study was to explore the experiences of in-patients affected by BU and leprosy in Liberia receiving integrated case management at Ganta rehabilitation centre.’, the quality of the study will be further strengthened if the specific research question(s) are stated. The research question(s) would help readers to know the specific issue(s) the authors will be addressing in their research. In addition, there is need for the authors to provide a clearer description of how integration was implemented in this setting, as this would be helpful for replicability of the intervention in a similar setting.

Although the authors did give an indication of their inclusion and exclusion criteria, they however did not state the reasons why these where adopted. The reasons for adopting these criteria need to be clearly stated.

There are no ethical or regulatory concerns as these have been adequately met.

Reviewer #2: “Minor Revision”

Reviewer #3: The methods sections needs to be organized properly. The design of the study was not stated but rather a description of what was done. Similarly, the authors did not clearly document how the participants were selected. 

describe the selection procedure

**Results**

-Does the analysis presented match the analysis plan?

-Are the results clearly and completely presented?

-Are the figures (Tables, Images) of sufficient quality for clarity?

Reviewer #1: The data analysis process is described in some detail, the authors mention they applied both inductive and deductive approaches to data analysis. From the article, it is not very clear why both approaches were adopted, what they mean how they were applied.

Reviewer #2: “Accept”.

Reviewer #3: Some areas of the results were methods. Line 221 to 225 should be part of the methods section

**Conclusions**

-Are the conclusions supported by the data presented?

-Are the limitations of analysis clearly described?

-Do the authors discuss how these data can be helpful to advance our understanding of the topic under study?

-Is public health relevance addressed?

Reviewer #1: The conclusions are supported by the data presented and are very much relevant to improve public health practice.

In the limitations section, the authors should consider mentioning the limitations of using a word processing software for transcription. Also, the authors may need to include comments on how rigour was ensured, such as trustworthiness, truth-value, applicability, consistency, and neutrality. 

The authors triangulated findings with health workers and their observations, did the patients themselves review and comment on the analysis as well?

Reviewer #2: “Minor Revision”

Reviewer #3: Conclusion was not very clear on the objectives but rather mixed with future research potentials. Its important that the authors conclude on the objectives of the study before making recommendations based on their findings

**Editorial and Data Presentation Modifications?**

Reviewer #1: Line 365 – 367 “A decent man, he’s working! The whole fingers 365 correct, the toes correct, everything on him is fine! No problem, you can’t even know he had leprosy."(Case 006, MBL, M, 21). It is good to see that the authors mentioned the positive impact of early treatment. In the discussion section, the authors might want to consider further emphasising that early diagnosis and treatment are crucial to minimizing morbidity, costs and prevent long-term disability in both leprosy and buruli ulcer patients. This could be a key driver for country programmes to conduct integrated active case finding interventions.

Line 133 ‘People with leprosy receive one year of multidrug therapy (rifampicin, clofazimine and dapsone)’. this statement is unclear as leprosy patients were not classified, and not all leprosy patients receive treatment for 12 months. Multibacillary (MB) leprosy patients are treated for 12 months, while paucibacillary (PB) leprosy patients are treated for 6 months. (see: http://www.searo.who.int/entity/global_leprosy_programme/approved-guidelines-leprosy-executives-summary.pdf?ua=1)

The authors may wish to consider including a table that highlights themes, subthemes etc. as this would provide a succinct summary of the findings and how the themes interact with each other.

Reviewer #2: “Minor Revision”

Reviewer #3: The methods sections needs to be organized and copyediting done before publishing. An interesting work

**Summary and General Comments**

Reviewer #1: This manuscript addresses a very important and relevant topic about which little has been published in the literature. This manuscript has great strength as it highlights the experiences of people affected by leprosy and Buruli ulcer of integrated case management in a resource constrained setting, pointing out implementation challenges control programmes applying integrated approaches could be confronted with in similar setting. Also, the use of multiple data sources (Interviews, observation, conversations with health workers, field notes) further enriched the quality of data.

Reviewer #2: The authors have done a good job by investigating the perception of patients, which has contributed to knowledge by being the first study, but reporting their findings seems to be a bit difficult. A review of their manuscript by someone proficient in English language would have helped.

Reviewer #3: Overall, an interesting work but some areas need to be worked on and copyeding is required to make the language ease to read. Kindly ensure that abstract brings out the key issues but in the current state it is not the best. Especially the introduction and the conclusion

PLOS authors have the option to publish the peer review history of their article (what does this mean?). If published, this will include your full peer review and any attached files.

Reviewer #1: No

Reviewer #2: Yes: Dr Vincent Pam Gyang

Reviewer #3: Yes: Ernest Kenu

---

## [Decision Letter · Decision Letter 1]

6 Jan 2020

Dear Dr Prochazka,

We are pleased to inform you that your manuscript, "“Buruli ulcer and leprosy, they are intertwined”: patient experiences of integrated case management of skin neglected tropical diseases in Liberia", has been editorially accepted for publication at PLOS Neglected Tropical Diseases.

Before your manuscript can be formally accepted and sent to production you will need to complete our formatting changes, which you will receive in a follow up email. Please note: your manuscript will not be scheduled for publication until you have made the required changes.

IMPORTANT NOTES

* Copyediting and Author Proofs: To ensure prompt publication, your manuscript will NOT be subject to detailed copyediting and you will NOT receive a typeset proof for review. The corresponding author will have one final opportunity to correct any errors when sent the requests mentioned above. Please review this version of your manuscript for any errors.

* If you or your institution will be <gwmw class="ginger-module-highlighter-mistake-type-3" id="gwmw-15781339505706902825725">preparing</gwmw> press materials for this manuscript, please inform our press team in advance at plosntds@plos.org. If you need to know your paper's publication date for media purposes, you must coordinate with our press team, and your manuscript will remain under a strict press embargo until the publication date and time. PLOS NTDs may choose to issue a press release for your article. If there is anything that the journal should know, please get in touch.

*Now that your manuscript has been provisionally accepted, please log into EM and update your profile. Go to <gwmw class="ginger-module-highlighter-mistake-type-3" id="gwmw-15781339548808726861237">http://www.editorialmanager.com/pntd, log</gwmw> in, and click on the "Update My Information" link at the top of the page. Please update your user information to ensure an efficient production and billing process.

*Note to LaTeX users only - Our staff will ask you to upload a TEX file in addition to the PDF before the paper can be sent to typesetting, so please carefully review our Latex Guidelines [http://www.plosntds.org/static/latexGuidelines.action] in the meantime.

Best regards,

Uwem Friday Ekpo, PhD

Associate Editor

Gerd Pluschke

Deputy Editor

Reviewer's Responses to Questions

**Key Review Criteria Required for Acceptance?**

**Methods**

-Are the objectives of the study clearly articulated with a clear testable hypothesis stated?

-Is the study design appropriate to address the stated objectives?

-Is the population clearly described and appropriate for the hypothesis being tested?

-Is the sample size sufficient to ensure adequate power to address the hypothesis being tested?

-Were correct statistical analysis used to support conclusions?

-Are there concerns about ethical or regulatory requirements being met?

Reviewer #1: The overall aim of the study is clear. Study design chosen is appropriate for the study undertaken and the thematic analysis approach employed for data analysis is very well described.

Reviewer #2: Accept

Reviewer #3: (No Response)

**Results**

-Does the analysis presented match the analysis plan?

-Are the results clearly and completely presented?

-Are the figures (Tables, Images) of sufficient quality for clarity?

Reviewer #1: The results are presented clearly and the authors' arguments can be easily followed.

Reviewer #2: Accept

Reviewer #3: (No Response)

**Conclusions**

-Are the conclusions supported by the data presented?

-Are the limitations of analysis clearly described?

-Do the authors discuss how these data can be helpful to advance our understanding of the topic under study?

-Is public health relevance addressed?

Reviewer #1: Conclusions are supported by the data presented and key study limitations have been described.

Reviewer #2: Accept

Reviewer #3: (No Response)

**Editorial and Data Presentation Modifications?**

Reviewer #1: Accept

Reviewer #2: Accept

Reviewer #3: (No Response)

**Summary and General Comments**

Reviewer #1: This research would add value to the current discourse on how to promote integrated services for NTDs, particularly skin NTDs in resource poor settings.

Reviewer #2: The revised manuscript has addressed the queries raised during <gwmw class="ginger-module-highlighter-mistake-type-3" id="gwmw-15781339895902517685965">review</gwmw>. This work has significance in the field of skin NTDs and could stimulate similar studies in different settings.

Under "Discussion" line 452; the authors need to delete one "these shared social experiences", because it was repeated.

Reviewer #3: (No Response)

PLOS authors have the option to publish the peer review history of their article (what does this mean?). If published, this will include your full peer review and any attached files.

If you choose “no”, your identity will remain <gwmw class="ginger-module-highlighter-mistake-type-3" id="gwmw-15781339939351116937563">anonymous but</gwmw> your review may still be made public.

Reviewer #1: No

Reviewer #2: Yes: Dr Vincent Pam Gyang

Reviewer #3: Yes: Dr Ernest Kenu

<gdiv></gdiv>

---

## [Editor Report · Acceptance letter]

22 Jan 2020

Dear Dr Prochazka,

We are delighted to inform you that your manuscript, "“Buruli ulcer and leprosy, they are intertwined”: patient experiences of integrated case management of skin neglected tropical diseases in Liberia," has been formally accepted for publication in PLOS Neglected Tropical Diseases.

Best regards,

Serap Aksoy

Editor-in-Chief

Shaden Kamhawi

Editor-in-Chief
